# Finite element analysis of the plantar support for the medial longitudinal arch with flexible flatfoot

Xiao Long[1☯], Xiangyu Du[1☯], Chengjie Yuan[2☯], Jian Xu[2], Tao Liu[3], Yijun Zhang[1]*

1 Department of Orthopedics, The First Affiliated Hospital, Zhejiang University School of Medicine, Hangzhou, P. R. China, 2 Department of Orthopedic Surgery, The First Affiliated Hospital, Zhejiang University School of Medicine, Hangzhou, P. R. China, 3 State Key Laboratory of Fluid Power and Mechatronic Systems, Zhejiang University School of Mechanical Engineering, Hangzhou, P. R. China

☯ These authors contributed equally to this work.
* zhangyijun87@zju.edu.cn

**Data Availability Statement:** All relevant data are within the paper and its Supporting Information files.

**Funding:** The author(s) received no specific funding for this work.

## Abstract

### Purpose

The present study is to explore the appropriate plantar support force for its effect on improving the collapse of the medial longitudinal arch with flexible flatfoot.

### Methods

A finite element model with the plantar fascia attenuation was constructed simulating as flexible flatfoot. The appropriate plantar support force was evaluated. The equivalent stress of the articular surface of the joints in the medial longitudinal arch and the maximum principal stress of the ligaments around the ankle were obtained.

### Results

The height fall is smaller when applying 15% of body-weight-bearing force as the plantar support for the medial longitudinal arch compared with 10% of the body-weight-bearing while 20% of body-weight-bearing force is over plantar support. The equivalent stress on the articular surface of each joint is smallest when applying 15% of body-weight-bearing force compared with 10% or 20% of the body-weight-bearing force. The maximum principal stress of the anterior talofibular ligament is decreased while other ligaments increased when the plantar fascia attenuation under loading. The maximum principal stress of the tibiocalcaneal ligament and the posterior tibiotalar ligament are decreasing while other ligaments increased with the force increasing gradually.

### Conclusions

Applying 15% of body-weight-bearing to the sole of the foot can restore the height fall of the medial longitudinal arch, and relieve the equivalent articular stress of the talonavicular joint and the talocalcaneal joint as well as the tension stress of the tibiocalcaneal ligament and the posterior tibiotalar ligament.

**Competing interests:** The authors have declared that no competing interests exist.

## Introduction

The treatment of symptomatic flexible flatfoot deformity can be roughly divided into non-invasive conservative rehabilitation and invasive surgical methods. Surgical treatment should only be considered after conservative treatments have failed. The standard conservative treatment for symptomatic flexible flatfoot deformity is orthotic therapy with the aid of insoles or ankle foot orthoses [1,2]. Foot orthoses are often used to correct altered gait pattern [3,4]. Previous cadaver experimental studies have evaluated the effectiveness of orthoses based on the load response of the tarsal bones [5,6] and reported improved hindfoot alignment in flatfoot deformity. Recent in vivo studies have been conducted by using video images or markers for motion analysis, but failed to demonstrate any beneficial effects of orthoses [7,8].

However, other studies reported that the foot orthosis as plantar support for medial longitudinal arch is an effective treatment for joints motion control, plantar pressure reduction and re-distribution in patients with flexible flatfoot deformity [3,4]. But the reported effectiveness has varied [7–10] for improper loading and it is still a controversial issue. Few studies have provided scientific evidence of applying proper loading insoles for flexible flatfoot deformity [4]. These study approaches require high financial investments in measurement equipment, as well as a meticulous control over the study samples that guarantees the biomechanical characteristics of the tissue [11].

An alternative approach nowadays accepted by clinicians and biomedical engineers is finite element (FE) modeling. This computational methodology allows the design of complex models that adequately represent the biomechanics of the human foot [12–16]. These models are considered as a valid alternative since researchers can include variations and loads over virtual structures that cannot be easily considered when using real tissue [17,18]. Of course, their validity depends on the correct design of physiological structures and the realistic modeling of the mechanical tissue properties [12,19–21]. Recent study had quantitative estimates of internal foot mechanics under various orthosis designs with flatfoot by a finite element study [22]. Zhang et al had analyzed the main soft tissue stress associated with flexible flatfoot deformity through a finite element study [23].

Therefore, in the present study, we will identify the body weight ratio that best supports the medial longitudinal arch using a finite element model with the plantar fascia attenuation simulating flexible flatfoot deformity. The hypothesis is that the appropriate plantar support force could improve the collapse of the medial longitudinal arch and alleviate the equivalent stress of each joint and the maximum principal stress of the ligaments around the ankle. Based on these results we will further design a novel air bladder inflation insole made with proper body loading to correct flexible flatfoot deformity as conservative treatment in clinical.

## Methods

### Finite element model design

The present study is based on the model proposed by Morales Orcajo et al [11,24]. The model reconstructs a normal human unloaded foot, based on tomography images (radiographs to 0.6 mm/slide) acquired from the right foot of a healthy male aged 35 years, with a height of 176 cm and a weight of 72 kg. The recruitment period for this study was from May 2$^{rd}$, 2021 to September 19$^{th}$, 2021. This study was conducted in accordance with ethical principles of research and was approved by the Medical Ethics Committee of the First Affiliated Hospital, Zhejiang University School of Medicine. The volunteer signed an informed consent form for the experimental protocol and purpose. The segmentation and tissue reconstruction were performed using MIMICS 20 software (Materialise, Leuven, Belgium). The Geometry processing

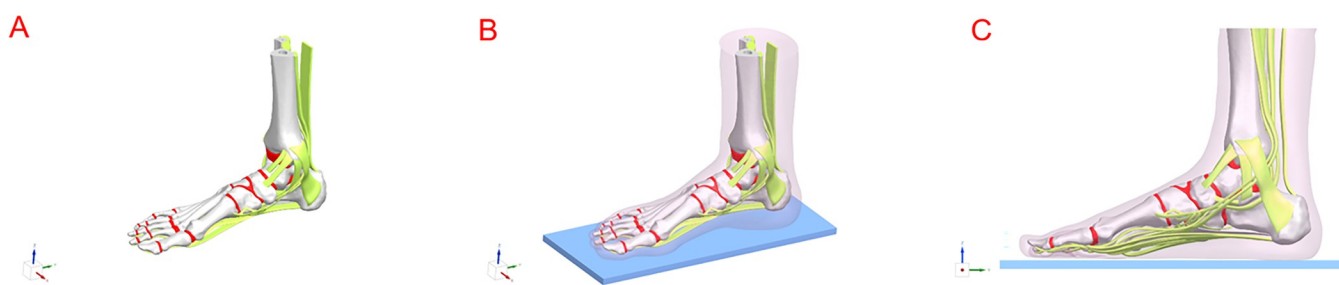

**Fig 1.** (A) 3D model reconstructed from CT images Oblique (B) and sagittal view (C) of the materialized model.

was performed using Geomagic Studio 2014 software (3D Systems, South Carolina, United States). The model refinement for further cutting and assembling was performed using Unigraphics NX 1911 software (Siemens, Munich, Germany). The model includes the bones and the cartilage morphology. The PF, SL, tendons, ligaments and fat pad were included based on anatomical images taken from atlases and cadaver dissection, under the advice of specialist foot and ankle surgeons. These tissues are fundamental for an adequate analysis of adult acquired flatfoot deformity (AAFD) development. In order to simulate the situation of standing on the ground, we designed a rectangular parallelepiped model larger than the sole surface of the foot with Unigraphics NX 1911 software. The sole plate size is about 300*120*12mm. The sole plate and the ground are relatively fixed, and there is no sliding (See Fig 1).

## Meshing

The model's meshing was performed using Ansys Workbench 2019 software (Canonsburg, Pennsylvania, United States), generating 28 bone pieces, 26 cartilage segments, 6 tendons, 7 ligaments, the plantar fascia and fat pad (See Fig 2A and 2B). The tetrahedral mesh of soft tissue that was generated is shown in Fig 2C as an example. A trialerror approach was employed to optimize the mesh size of each segment, following the recommendations of Burkhart et al [25] who suggest that in all the parameters measured. The equilibrium was found with 1,401,813 linear tetrahedral elements (C3D4) with element sizes as follows: 1 mm for the smallest cartilages between phalanges, 2 mm for the phalanges, the thinnest ligaments and the rest of the cartilages, 3 mm for the metatarsals and the rest of the tendons, and 5 mm for the large bones in the hindfoot. The solution time for the model was CPU time 3.5h (CPU- Intel Xeon Gold 6230 40 cores Memory 192G).

## Boundary condition

The model reconstructs a non-weight-bearing foot (unloaded), thus an initial simulation to obtain a loading position was performed. The model was simulated including all the tissues

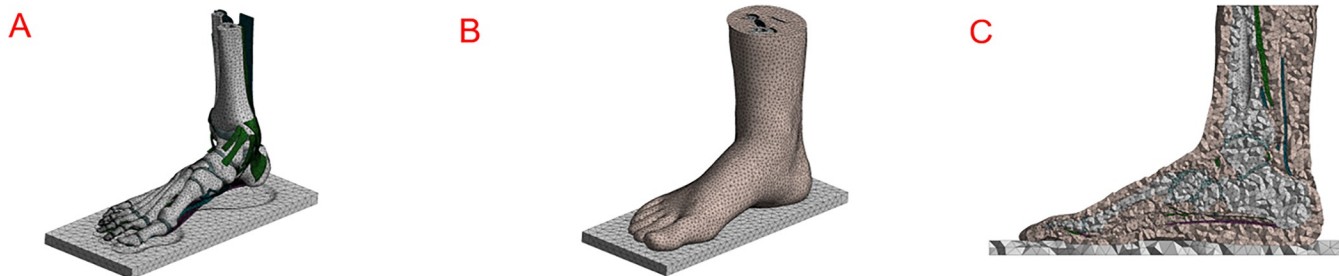

**Fig 2.** The meshes of bones (A) and soft tissue (B) and sole plate (C) The cross-section view of the soft tissue mesh.

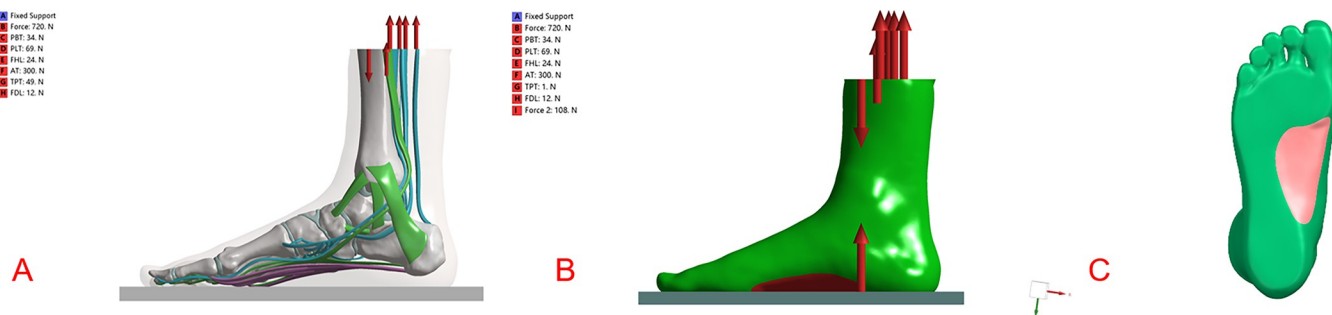

**Fig 3.** The model was simulated using a load that represents full-weight-bearing and applying traction force to the related tendons that represents the dynamic stabilizer for the ankle (A). Applying force to the sole to simulate plantar support for the medial longitudinal arch (B). The red area is the plantar support area (C).

using a 720N load that represents the full-weight-bearing of an adult person about 72Kg, leaning on one foot. This condition emulates a traditional AAFD diagnostic assessment scenario.

The load was introduced in a descending vertical direction, with 10 degrees of inclination (distributed in the zone of contact Tibia-Astragalus (90%) and Fibula-Astragalus (10%) [11]. The simulations were performed by maintaining fixed nodes at the lower part of the calcaneus and blocking the Z-axis displacement (vertical) of the lower nodes of the metatarsals. This was done in order to simulate the ground effect when an adult person is leaning on one foot. Meanwhile, applying traction force to peroneus longus tendon, peroneus brevis tendon, flexor longus tendon, Achilles tendon, posterior tibial tendon, flexor digitorum longus tendon as reported in the Arangio and Salathe study (set peroneal longus tendon 69N, peroneal brevis tendon 34N, flexor longus tendon 24N, Achilles tendon 300N, posterior tibial tendon 49N, flexor digitorum longus tendon 12N) for simulating dynamic stabilizer of the plantar arch (See Fig 3A) [26].

### Tissue biomechanical properties

The model includes the plantar fascia, spring ligament, the anterior talofibular ligament, the calcaneofibular ligament, the posterior talofibular ligament, the deltoid ligament, and the anterior and posterior tibiofibular syndesmosis as well as major tendons including Achilles tendon, peroneal brevis tendon, posterior tibial tendon, flexor digitorum longus tendon, and flexor digitorum longus tendon in appropriate anatomical positions, and fat pad. These tissue models were considered such as an elastic-linear material, using biomechanical properties reported in the literature: bone (E = 17,000 MPa, v = 0.3), ligaments (E = 260 MPa, v = 0.4) and plantar fascia (E = 350 MPa, v = 0.4), fat pad (E = 1.0MPa, v = 0.45), E being Young's modulus and v Poisson's ratio [27,28]. The tendons and cartilage were modeled as hyper-elastic materials (Ogden model), using the parameters (tendons: E = 1200MPa, v = 0.4; cartilage: E = 10MPa, v = 0.49) used in specialized articles [29,30]. The Ogden model describes the hyperelastic behavior of rubber-like materials. Its strain energy density function U is:

$$U = \frac{\mu}{a^2} \left( \lambda_1^a + \lambda_2^\alpha + \lambda_3^a - 3 \right) + \frac{1}{D}(J-1)^2$$

where the initial shear modulus μ = 4.4, the strain hardening exponent α = 2 and the compressibility parameter D = 0.45.

In our model, for the ankle joint, the automated surface-to-surface contact option in Ansys Workbench was used to simulate the frictionless contact relationship between articular surfaces. Intertarsal joints, tarsometatarsal joints, metatarsophalangeal joints and intermetatarsal

joints were assumed to exhibit only small movements in the standing condition and were therefore simplified by connecting the articular surfaces with solid elements comprising cartilage stiffness. The tissue to bone contact property was bonded contact relationship and the tissue to ligament contact property was no separation contact relationship which means separation of surfaces in contact is not allowed but small amounts of frictionless sliding can occur along contact surfaces. The tissue failures or attenuation applied to simulate AAFD development were performed applying the Isotropic Hardening theory that generates a progressive tissue attenuation [31].

## Applying optimal plantar support force for finite element model

We applied supporting load on the sole of the foot, respectively 10% (72N), 15% (108N), 20% (144N) of the body-weight-bearing as simulating the support effect of the insole on the arch of the foot, and evaluated the appropriate arch support force. The supporting load is uniformly distributed in the foot plantar support area (see Fig 3B and 3C).

## Evaluation criteria and simulation conditions

To determine the plantar arch height and the relative contribution of each tissue, we calculated the difference between each performed simulation and the results of the model in normal load conditions. To quantify the quasi-clinical deformation values of the model and obtain a relative comparison of each analyzed tissue, we performed a simulation maintaining the bones, cartilage, ligaments and tendons, following the methodology proposed by Tao et al [28] for a tissue experimental test using cadaver models. In this way, the quasi-clinical possible deformation of our model was obtained. The height fall of the medial longitudinal arch was evaluated following the displacement of the lower part of the head of the talus, navicular, midpoint of medial cuneiform and the first metatarsal. In order to determine the biomechanical contribution of each tissue, the simulations were carried out maintaining and weakening each one of the evaluated tissues. Although damaged tissues continue working after an injury, herein we wanted to identify how important each tissue is to maintain the foot arch in a normal position. The normal standing load was considered as a reference standard. Subsequently, the flexible flatfoot was simulated with the PF attenuation under full weight-bearing condition followed by applying force to the sole to simulate the plantar support for the medial longitudinal arch. The variation of the height fall of the medial longitudinal arch, equivalent stress on the articular surface of each joint in the medial longitudinal arch and maximum principal stress of the ligaments around the ankle were evaluated.

## Validation of the foot finite element model

The model constructed in this study was validated following the recommendations of Tao et al [32], measuring some anatomical parameters from the sagittal view under different loading conditions (non-weight bearing and normal standing weight bearing). The changes of these anatomical points allow us to compare the vertical displacements visible in radiographic images of a normal foot with respect to the finite element model predictions. We measured the vertical distance of the highest point of the Talus (TAL), the Navicular (NAV), the middle of the Cuneiform (CUN), and the highest point of the first metatarsal head (MTH1), as can be seen in Fig 4. By measuring the vertical displacement change of these points in twelve radiographic images (under non-weight-bearing and normal standing weight-bearing condition) of six patients' right foot, the average value and standard deviation were obtained, and the model prediction results were objectively compared. All the six patients signed an informed consent form for the experimental protocol and purpose. The demographic details of the six patients is

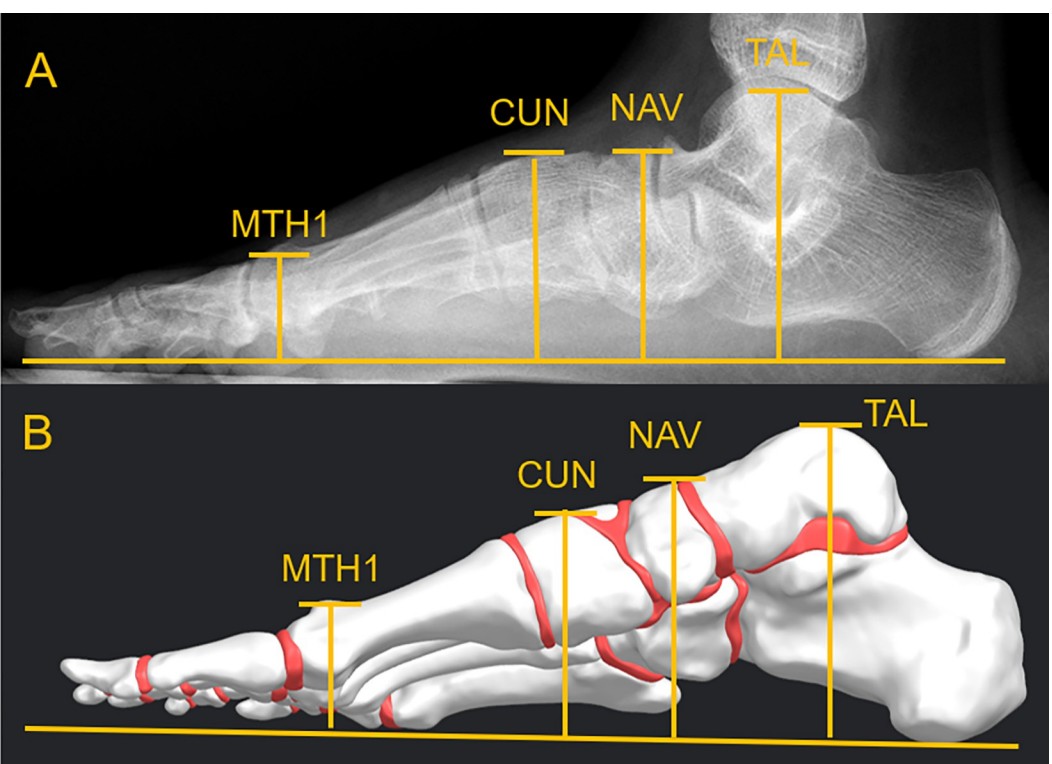

**Fig 4.** The height of the medial longitudinal arch (A Real patient X-rays) (B Model simulation).

shown in Table 1. The acceptable data difference between the predicted model and patient measurement according to the study reported was within ±0.25 mm for all cases [32].

## Results

### Validation of the finite element model

Results of the validation process can be seen in Table 2. The model generates a plantar arch fall similar to a healthy patient in a loading test, simulating all the tissues in normal and functional conditions. The evaluation was performed observing the foot anatomy in a sagittal view under two conditions: no weight-bearing (without soft tissue tension) and full weight-bearing (soft tissue tension under normal conditions).

### Biomechanical evaluation of the finite element model with the PF attenuation

The height fall of the medial longitudinal arch on the sagittal plane and the talus medial displacement on the frontal plane.

**Table 1. The demographic details of the six patients.**

| No | Male/Female | Age(y) | Weight(kg) | Height(cm) |
|----|-------------|--------|------------|------------|
| 1 | Male | 35 | 71.5 | 170.2 |
| 2 | Female | 46 | 73.2 | 175.4 |
| 3 | Female | 38 | 68.4 | 165.5 |
| 4 | Male | 27 | 70.2 | 168.7 |
| 5 | Male | 26 | 72.0 | 172.6 |
| 6 | Male | 40 | 75.1 | 178.1 |

**Table 2. Results of the validation process.** The values correspond to the difference between the measured distance from each point to the ground, under two different conditions: No weight-bearing and full weight-bearing.

| Reference point | Model prediction (mm) | Patient average (mm) | Patient std. deviation |
|---|---|---|---|
| TAL | 0.292 | 0.290 | 0.03 |
| NAV | 0.330 | 0.288 | 0.06 |
| CUN | 0.324 | 0.265 | 0.12 |
| MTH1 | 0.056 | 0.089 | 0.09 |

The height fall variation of the medial longitudinal arch on the sagittal plane and the talus medial displacement on the frontal plane are showed in Table 3. The results showed that the height fall is smaller when applying about 15% of body-weight-bearing force as the plantar support for the medial longitudinal arch compared with 10% of the body-weight-bearing. However, when 20% of body-weight-bearing force was applied, we found that the height fall variation of the first metatarsal was increased, indicating over plantar support for the medial longitudinal arch. Moreover, the talus medial displacement is decreased as the plantar support increasing gradually.

Equivalent stress on the articular surface of each joint of the medial longitudinal arch.

The equivalent stress variation on the articular surface of each joint of the medial longitudinal arch of the foot is showed in Table 4. The results showed that the equivalent stress on the articular surface of each joint is smallest when applying about 15% of body-weight-bearing force as the plantar support for the medial longitudinal arch compared with 10% or 20% of the body-weight-bearing force.

## Maximum principal stress variation of the ligaments around the ankle

The maximum principal stress variation of the ligaments around the ankle is showed in Tables 5 and 6. The results showed that the maximum principal stress of the anterior talofibular ligament is decreased while all the other ligaments increased when the PF attenuation under loading. When applying the plantar support for the medial longitudinal arch, we found that the maximum principal stress of the tibiocalcaneal ligament and the posterior tibiotalar ligament are decreasing while the remaining ligaments increased with the force increasing gradually.

## Discussion

The present study was to determine the appropriate support force of the individualized insole and analyze its corrective effect on flexible flatfoot by a three dimensional finite element

**Table 3. The height fall of the medial longitudinal arch on the sagittal plane and the talus medial displacement on the frontal plane in normal foot and flexible flat-foot model under loading and plantar supporting.**

| Group | Height fall of the medial longitudinal arch (mm) | | | | Talus medial displacement (mm) |
|---|---|---|---|---|---|
| | TAL | NAV | CUM | MTH1 | |
| Normal | 1.035 | 1.358 | 1.004 | 0.155 | 1.004 |
| PF attenuation | 1.351 | 1.816 | 1.325 | 0.258 | 1.108 |
| Applying force (10%) | 1.119 | 1.439 | 0.806 | 0.086 | 0.930 |
| Applying force (15%) | 1.005 | 1.256 | 0.554 | 0.002 | 0.844 |
| Applying force (20%) | 0.886 | 1.062 | 0.287 | -0.087 | 0.753 |

"-" means the height was increased.

**Table 4. Equivalent stress on the articular surface of each joint in the medial longitudinal arch in normal foot and flexible flatfoot model under loading.**

| Group | Equivalent stress on the articular surface of each joint (MPa) | | | |
|---|---|---|---|---|
| | Talocalcaneal joint | Talonavicular joint | Medial cuneonavicular joint | First tarsometatarsal joint |
| Normal | 0.186 | 0.382 | 0.316 | 0.105 |
| PF attenuation | 0.217 | 0.429 | 0.332 | 0.115 |
| Applying force(10%) | 0.201 | 0.403 | 0.324 | 0.112 |
| Applying force(15%) | 0.151 | 0.359 | 0.309 | 0.099 |
| Applying force(20%) | 0.195 | 0.398 | 0.331 | 0.122 |

model. We applied the force with 10%, 15% and 20% of the body-weight-bearing respectively simulating as the insole support for flexible flatfoot model, and evaluated its influence on the height fall variation of the medial longitudinal arch, the equivalent stress of the articular surface of each joint and the maximum principal stress of the ligaments around the ankle. The results showed that the height fall is smaller when applying about 15% of body-weight-bearing force as the plantar support for the medial longitudinal arch compared with 10% of the body-weight-bearing. However, when 20% of body-weight-bearing force was applied, we found that the height of the first metatarsal was not dropped but elevated, indicating over plantar support for the medial longitudinal arch. Zhang et al evaluated the load response difference between the flexible flatfoot and healthy foot at the medial longitudinal arch joints and reported that the flexible flatfoot dorsiflexed more in the talocalcaneal joint, the medial cuneonavicular joint and the first tarsometatarsal joint compared with the healthy foot [33]. Therefore, we believe that applying an appropriate support to the sole of the foot can decrease dorsiflexion of the medial longitudinal arch joints and promote the recovery of the medial longitudinal arch collapse. Chen et al [7] and Kulcu et al [8] performed gait analysis of flatfoot by an optical surface marking system and studied the effect of insoles on the correction of flatfoot deformity. However, neither of these two studies had studied the proper plantar support and proved the effectiveness of orthotic insole support.

Our study explored the conservative treatment of patients with flexible flatfoot deformity by using custom-made insole based on partial body-weight-bearing as the plantar support force. The results show that if 10% of the body-weight-bearing is applied as the plantar support force, the medial longitudinal arch of the flatfoot will be insufficiently supported, while if 20% of the human body-weight-bearing is applied, the medial longitudinal arch of the flatfoot will be over supported. Therefore, 15% of body-weight-bearing applied as the plantar support force of the medial longitudinal arch of the flatfoot is effective, which can improve the collapse of the arch and restore to the variation of the medial longitudinal arch after the healthy foot is loaded. This is consistent with the previous results reported by several authors that orthotic insoles had effect on flexible flatfoot. Lee et al [20] found that orthotic insoles can increase the support of the medial longitudinal arch of the foot during the gait cycle, thereby reduce stress

**Table 5. Maximum principal stress variation of the lateral ankle ligaments in normal foot and flexible flatfoot model under loading.**

| Group | Maximum principal stress variation of the lateral ligaments (Mpa) | | |
|---|---|---|---|
| | Anterior talofibular ligament | Calcaneofibular ligament | Posterior talofibular ligament |
| Normal | 5.070 | 27.634 | 6.588 |
| PF attenuation | 4.863 | 28.827 | 6.953 |
| Applying force(10%) | 4.878 | 29.109 | 7.414 |
| Applying force(15%) | 4.885 | 29.246 | 7.638 |
| Applying force(20%) | 4.893 | 29.391 | 7.875 |

**Table 6. Maximum principal stress variation of the medial ankle ligaments in normal foot and flexible flatfoot model under loading.**

| Group | Maximum principal stress variation of the medial ligaments (Mpa) | | | |
|---|---|---|---|---|
| | Anterior tibiotalar ligament | Tibiocalcaneal ligament | Tibionavicular ligament | Posterior tibiotalar ligament |
| Normal | 11.818 | 4.736 | 8.541 | 1.905 |
| PF attenuation | 11.824 | 4.789 | 8.763 | 2.013 |
| Applying force(10%) | 12.118 | 4.656 | 8.951 | 1.982 |
| Applying force(15%) | 12.261 | 4.592 | 9.042 | 1.970 |
| Applying force(20%) | 12.412 | 4.523 | 9.138 | 1.958 |

due to excessive contraction of the intrinsic and extrinsic muscles maintaining the medical longitudinal arch and improving the balance function of the ankle. However, they had not study the proper support loading of sole for flexible flatfoot study.

To our knowledge, it is the first study to identify appropriate body weight ratio that best supports the medial longitudinal arch and analyze the equivalent stress variation of the articular surface of each joint in the medial longitudinal arch and the maximum principal stress of the ligaments around the ankle before and after loading. The results showed that the equivalent stress of the articular surface of each joint in the medial longitudinal arch increases when the PF attenuation. With the gradual increase of the force applied on the sole, we found that the equivalent stress variation of the articular surface of each joint decreases first and then increases. When applying about 15% of body-weight-bearing force as the plantar support for the medial longitudinal arch, the equivalent stress is the smallest compared with 10% or 20% of body-weight-bearing force. The probable reason is that vertical body-weight-bearing force transfers medially with the hindfoot excessive valgus movement and the medial longitudinal arch collapse in flexible flatfoot, thus increasing the equivalent stress of the articular surface of the joints. The appropriate plantar support could counteract the vertical force. In addition, we found that the most important effect of plantar support on the medial longitudinal arch of the foot is the talonavicular joint and the talocalcaneal joint due to their large variation. Restricting the over mobility of the talonavicular and talocalcaneal joints may be useful for correcting the medial longitudinal arch and treating the flexible flatfoot deformity.

Regarding the maximum principal stress of the ligaments around the ankle, we found that the anterior talofibular ligament is decreased while the remaining ligaments increased when the PF attenuation under loading. Moreover, the tibiocalcaneal ligament and the posterior tibiotalar ligament are decreased while other ligaments increased with the force as the plantar support for the medial longitudinal arch increasing gradually. Therefore, we believe that plantar support has mainly effect on the stress relieving of the tibiocalcaneal ligament and the posterior tibiotalar ligament. Zhang et al had confirmed that the eversion of the talocalcaneal joint had significant influence on the medial longitudinal arch from non- to full-body-weight bearing condition [33]. Kitaoka et al had also reported that much of the pes planus malalignment was caused by deformation at the talocalcaneal joint [34]. So the reason for the ligaments stress variation is probably correlated with correction of the talocalcaneal joint eversion in flexible flatfoot deformity.

The present study also had some limitations. First, the material properties used for the bony and ligamentous structures in this study were obtained from previous studies, which could underestimate the prediction accuracy of the model. Second, the single-subject model design for the EF analysis was used in this study, which could not account for population variances, such as arch height, body weight, foot stiffness, and foot symptoms. Further work should be

conducted to consider the patient variances. Thirdly, only static forces were adopted to simulate plantar support; dynamic foot-insole pressures should be considered to improve the EF analysis accuracy, which could evaluate dynamic plantar support for the medial longitudinal arch during gait cycle. Forth, the vertical loading of the ankle in a neutral position can only simulate a human standing state and the plantar supporting force is limited to quasi-static condition. Further work should be conducted to investigate the supporting force variation in more dynamic activities, which is helpful to gain a deeper understanding of the effect of individualized insoles on patients with flexible flatfoot dynamically.

## Conclusions

In this study, the appropriate support force of the individualized insole and its corrective effect on flexible flatfoot were investigated by constructing a three-dimensional finite element model. The results indicated that applying about 15% of the human body-weight-bearing as plantar support to the medial sole of the foot can restore the height fall of the medial longitudinal arch of the foot, and relieve the equivalent articular stress of the talonavicular joint and the talocalcaneal joint as well as the tension stress of the tibiocalcaneal ligament and the posterior tibiotalar ligament, thus correcting flexible flatfoot deformity as conservative treatment. The results of this study could provide the theory for a novel individualized air bladder inflation orthotic insole design in flexible flatfoot conservative treatment in the future.

## Acknowledgments

We thank Mechanical Engineer XX for help with finite element model construction.

## Author Contributions

**Data curation:** Tao Liu.

**Methodology:** Jian Xu.

**Project administration:** Yijun Zhang.

**Validation:** Tao Liu.

**Writing – original draft:** Xiao Long, Xiangyu Du.

**Writing – review & editing:** Chengjie Yuan.

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
