## [Decision Letter · Decision Letter 0]

10 Jul 2024

PONE-D-24-19378Finite element analysis of the plantar support for the medial longitudinal arch with flexible flatfootPLOS ONE

Dear Dr. Zhang,

Thank you for submitting your manuscript to PLOS ONE. After careful consideration, we feel that it has merit but does not fully meet PLOS ONE’s publication criteria as it currently stands. Therefore, we invite you to submit a revised version of the manuscript that addresses the points raised during the review process.

We look forward to receiving your revised manuscript.

Kind regards,

Kentaro Amaha

Academic Editor

PLOS ONE

3. PLOS requires an ORCID iD for the corresponding author in Editorial Manager on papers submitted after December 6th, 2016. Please ensure that you have an ORCID iD and that it is validated in Editorial Manager. To do this, go to ‘Update my Information’ (in the upper left-hand corner of the main menu), and click on the Fetch/Validate link next to the ORCID field. This will take you to the ORCID site and allow you to create a new iD or authenticate a pre-existing iD in Editorial Manager. Please see the following video for instructions on linking an ORCID iD to your Editorial Manager account: https://www.youtube.com/watch?v=_xcclfuvtxQ".

6. We note that Figure(s) 1, 2, 3 and 4 in your submission contain copyrighted images. All PLOS content is published under the Creative Commons Attribution License (CC BY 4.0), which means that the manuscript, images, and Supporting Information files will be freely available online, and any third party is permitted to access, download, copy, distribute, and use these materials in any way, even commercially, with proper attribution. For more information, see our copyright guidelines: http://journals.plos.org/plosone/s/licenses-and-copyright.

a. You may seek permission from the original copyright holder of Figure(s) 1, 2, 3 and 4 to publish the content specifically under the CC BY 4.0 license. 

Additional Editor Comments:

Please correct the reviewer's point. Also, the results of insoles on flat feet have been studied in many ways. Please describe in more detail what can be expected clinically from this study.

Reviewers' comments:

Reviewer's Responses to Questions

**Comments to the Author**

1. Is the manuscript technically sound, and do the data support the conclusions?

Reviewer #1: Yes

Reviewer #2: No

2. Has the statistical analysis been performed appropriately and rigorously? 

Reviewer #1: Yes

Reviewer #2: No

3. Have the authors made all data underlying the findings in their manuscript fully available?

Reviewer #1: Yes

Reviewer #2: No

4. Is the manuscript presented in an intelligible fashion and written in standard English?

Reviewer #1: Yes

Reviewer #2: No

5. Review Comments to the Author

Reviewer #1: Dear Author,

I would like to start my review by expressing my pleasure in examining your work. Your identification of the appropriate loading rate in your study is particularly important for conservative treatment.

Your work is a biomechanical study. Therefore, it would be more appropriate to discuss foot biomechanics and the problems caused by orthoses with improper loading rather than orthoses in the introduction section.

The objective of your study differs between the introduction and discussion sections. These parts should be consistent. Is your study about determining the shape of insoles to be made with proper body loading, or is it a biomechanical study in which you identify the body weight ratio that best supports the medial longitudinal arch? You need to clarify this.

It is very good that you have mentioned your limitations. However, it would be more informative and guiding if you also write down the reasons for these limitations.

Respectfully

Reviewer #2: This study found that applying 15% of body-weight-bearing to the sole of the foot can restore the height fall of the medial longitudinal arch, and relieve the equivalent articular stress of the talonavicular joint and the talocalcaneal joint as well as the tension stress of the tibiocalcaneal ligament and the posterior tibiotalar ligament. But this study is not novel enough to be accepted.

6. PLOS authors have the option to publish the peer review history of their article (what does this mean?). If published, this will include your full peer review and any attached files.

Reviewer #1: No

Reviewer #2: No

---

## [Author Response · Author response to Decision Letter 0]

7 Sep 2024

Reply to reviewers’ comments

Reviewer(s)' Comments to Author:

Reviewer #1: Dear Author,

I would like to start my review by expressing my pleasure in examining your work. Your identification of the appropriate loading rate in your study is particularly important for conservative treatment.

Your work is a biomechanical study. Therefore, it would be more appropriate to discuss foot biomechanics and the problems caused by orthoses with improper loading rather than orthoses in the introduction section.

The objective of your study differs between the introduction and discussion sections. These parts should be consistent. Is your study about determining the shape of insoles to be made with proper body loading, or is it a biomechanical study in which you identify the body weight ratio that best supports the medial longitudinal arch? You need to clarify this.

It is very good that you have mentioned your limitations. However, it would be more informative and guiding if you also write down the reasons for these limitations.

Respectfully

Reply: OK. We have revised in the text. See line 69-76 “Recent in vivo studies have been conducted by using video images or markers for motion analysis, but failed to demonstrate any beneficial effects of orthoses [7-8]. However, other studies reported that the foot orthosis as plantar support for medial longitudinal arch is an effective treatment for joints motion control, plantar pressure reduction and re-distribution in patients with flexible flatfoot deformity [3-4]. But the reported effectiveness has varied [7-10] for improper loading and it is still a controversial issue. Few studies have provided scientific evidence of applying proper loading insoles for flexible flatfoot deformity [4]”. See line 91-98 “Therefore, in the present study, we will identify the body weight ratio that best supports the medial longitudinal arch using a finite element model with the plantar fascia attenuation simulating flexible flatfoot deformity. The hypothesis is that the appropriate plantar support force could improve the collapse of the medial longitudinal arch and alleviate the equivalent stress of each joint and the maximum principal stress of the ligaments around the ankle. Based on these results we will further design a novel air bladder inflation insole made with proper body loading to correct flexible flatfoot deformity as conservative treatment in clinical.” See line 336-341 “However, neither of these two studies had studied the proper plantar support and proved the effectiveness of orthotic insole support.” See line 356-366 “However, they had not study the proper support loading of sole for flexible flatfoot study. To our knowledge, it is the first study to identify appropriate body weight ratio that best supports the medial longitudinal arch and analyze the equivalent stress variation of the articular surface of each joint in the medial longitudinal arch and the maximum principal stress of the ligaments around the ankle before and after loading.” See line 397-404 “Second, the single-subject model design for the EF analysis was used in this study, which could not account for population variances, such as arch height, body weight, foot stiffness, and foot symptoms. Further work should be conducted to consider the patient variances. Thirdly, only static forces were adopted to simulate plantar support; dynamic foot-insole pressures should be considered to improve the EF analysis accuracy, which could evaluate dynamic plantar support for the medial longitudinal arch during gait cycle.” See line 413-421 “The results indicated that applying about 15% of the human body-weight-bearing as plantar support to the medial sole of the foot can restore the height fall of the medial longitudinal arch of the foot, and relieve the equivalent articular stress of the talonavicular joint and the talocalcaneal joint as well as the tension stress of the tibiocalcaneal ligament and the posterior tibiotalar ligament, thus correcting flexible flatfoot deformity as conservative treatment. The results of this study could provide the theory for a novel individualized air bladder inflation orthotic insole design in flexible flatfoot conservative treatment in the future.”

Reviewer #2: This study found that applying 15% of body-weight-bearing to the sole of the foot can restore the height fall of the medial longitudinal arch, and relieve the equivalent articular stress of the talonavicular joint and the talocalcaneal joint as well as the tension stress of the tibiocalcaneal ligament and the posterior tibiotalar ligament. But this study is not novel enough to be accepted.

Reply: To our knowledge, it is the first study to identify appropriate body weight ratio that best supports the medial longitudinal arch and analyze the equivalent stress variation of the articular surface of each joint in the medial longitudinal arch and the maximum principal stress of the ligaments around the ankle before and after loading. The results indicated that applying about 15% of the human body-weight-bearing as plantar support to the medial sole of the foot can restore the height fall of the medial longitudinal arch of the foot, and relieve the equivalent articular stress of the talonavicular joint and the talocalcaneal joint as well as the tension stress of the tibiocalcaneal ligament and the posterior tibiotalar ligament, thus correcting flexible flatfoot deformity as conservative treatment. The results of this study could provide the theory for a novel individualized air bladder inflation orthotic insole design in flexible flatfoot conservative treatment in the future.

---

## [Editor Report · Decision Letter 1]

28 Oct 2024

Finite element analysis of the plantar support for the medial longitudinal arch with flexible flatfoot

PONE-D-24-19378R1

Dear Dr. Zhang,

We’re pleased to inform you that your manuscript has been judged scientifically suitable for publication and will be formally accepted for publication once it meets all outstanding technical requirements.

Kind regards,

Kentaro Amaha

Academic Editor

PLOS ONE

---

## [Editor Report · Acceptance letter]

1 Nov 2024

PONE-D-24-19378R1 

PLOS ONE

Dear Dr. Zhang, 

I'm pleased to inform you that your manuscript has been deemed suitable for publication in PLOS ONE. Congratulations! Your manuscript is now being handed over to our production team.

Kind regards, 

on behalf of

Dr. Kentaro Amaha 

Academic Editor

PLOS ONE